# CORRELATING CROSS-ITERATION NOISE FOR DP-SGD USING MODEL CURVATURE

## ABSTRACT

Differentially private stochastic gradient descent (DP-SGD) offers the promise of training deep learning models while mitigating many privacy risks. However, there is currently a large accuracy gap between DP-SGD and normal SGD training. This has resulted in different lines of research investigating orthogonal ways of improving privacy-preserving training. One such line of work, known as DP-MF, correlates the privacy noise across different iterations of stochastic gradient descent – allowing later iterations to cancel out some of the noise added to earlier iterations. In this paper, we study how to improve this noise correlation. We propose a technique called NoiseCurve that uses model curvature, estimated from public unlabeled data, to improve the quality of this cross-iteration noise correlation. Our experiments on various datasets, models, and privacy parameters show that the noise correlations computed by NoiseCurve offer consistent and significant improvements in accuracy over the correlation scheme used by DP-MF.

## 1 INTRODUCTION

Differential privacy (DP) (Dwork et al., 2006b) is a rigorous mathematical framework that limits the amount of personal information an attacker can infer from the output of an algorithm that processes confidential data. Differentially private stochastic gradient descent (DP-SGD, (Abadi et al., 2016)) is one of the most popular methods for training machine learning (ML) models with DP guarantees. DP-SGD differs from standard SGD in two important ways. First, the gradient of each sample within a batch is separately calculated and clipped by a constant $\zeta$ before being averaged. Second, *isotropic* Gaussian noise is added to the averaged (clipped) gradient in each iteration before model weights are updated. Although promising, the barrier to wide-spread adoption is the accuracy gap between models trained with DP-SGD and models trained without this privacy protection.

To improve the accuracy of DP-SGD, several recent works, which we call *DP-MF* methods, explored replacing independent Gaussian noise with noise that is **correlated across iterations**. The correlations are carefully designed so that noise in some iterations can partially cancel out noise used in other iterations, while providing the same level of privacy (Kairouz et al., 2021; Denisov et al., 2022; Choquette-Choo et al., 2023a). The noise correlation is controlled by a mixing matrix $C$ that is obtained by minimizing a heuristic objective function subject to privacy constraints. Existing DP-MF methods use a heuristic objective function which is a data-independent approximation of how the noise would cause parameter updates in DP-SGD to differ from ordinary SGD.

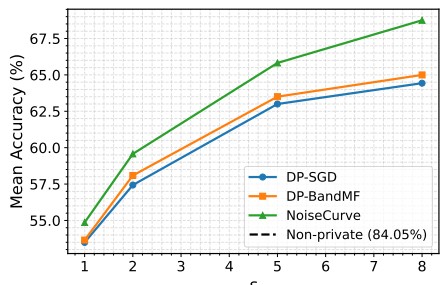

Figure 1: Test accuracy (averaged over 3 runs) on CIFAR-10 for $\delta = 10^{-5}$ as $\epsilon$ varies. Details in Section 5. Curvature is obtained in an unsupervised way from TinyImageNet.

In this paper, we make the critical observation that the noise added in each iteration causes the training trajectory of DP-SGD to diverge from ordinary SGD in *two* important ways: (1) as is widely recognized, it makes the gradient at that iteration noisy; (2) as a result, it alters the point at which the gradient of the *subsequent* iteration is computed — the change in this subsequent gradient is

therefore affected by the model Hessian (details are in Section 2). The current DP-MF objective function only considers the first error and *disregards the second*, most likely because it appears to be extremely data-dependent. However, we find that *some* curvature information, even an imperfect one derived from public datasets, can help find noise correlations that consider both errors and significantly improve accuracy of privacy-preserving model training (see Figure 1).

In this paper, we replace the DP-MF heuristic objective function with a different data-independent (heuristic) objective that better accounts for these two sources of errors. We do this by working out what the optimal objective would be in the idealized quadratic case with a perfectly known Hessian. It turns out the relevant curvature information is completely captured by the *Hessian eigenvalues*. We show how to estimate good representative curvature information, from unlabeled public datasets, that can be used in training non-convex models for which the Hessian (1) changes as weights get updated, (2) often has negative eigenvalues, and (3) is too expensive to compute. Our proposed approach, *NoiseCurve*, is a drop-in replacement for the objective function in the DP-MF framework. Below, we summarize our contributions:

1. **We propose a new objective that accounts for the interaction between model curvature and differentially private SGD with correlated noise.** The new objective is easy to incorporate into existing software packages because it is a drop-in replacement for the DP-MF objective.

2. **We address practical challenges to obtaining curvature information for deep learning models, without incurring additional privacy loss**. These include obtaining curvature information from unlabeled public data, and coping with the fact that neural network Hessians change during training and have negative eigenvalues. Our approach, called NoiseCurve, can improve the privacy-preserving model accuracy across a wide range of setups (e.g., Figure 1 shows an accuracy improvement of **1–4%** compared than DP-SGD and DP-BANDMF, a state-of-the-art (SOTA) DP-MF method, across all the privacy budgets we tested).

3. **We address scalability to models for which full eigenspectrum computation is infeasible.** When the model is too large, computing only the top-$k$ largest eigenvalues are feasible, with $k$ being in the order of hundreds to thousands. However, we observed that only using them is ineffective, as they provide a misleading curvature information (that gradients will always stay within a fixed $k$-dimensional space that is too small). While misleading curvature information does not affect privacy, it affects the quality of the resulting noise correlations. We propose an efficient method that combines eigenvalue estimation with implicit estimation of the *size* of the subspace the gradients are likely to belong to (roughly, the *number* of positive eigenvalues).

## 1.1 RELATED WORK

**Using Correlated Noise (DP-MF).** Cross-iteration correlated noise was first explored in DP-FTRL (Kairouz et al., 2021), which used tree aggregation. Later works (DP-MF) generated correlated noise through solving a matrix factorization problem (Denisov et al., 2022). Choquette-Choo et al. (2023b) extended it to multi-epoch training. Concurrently, Koloskova et al. (2023) used theoretical convergence analysis to propose rescaling different parts of their objective function. State-of-the-art accuracy in this line of work was achieved by DP-BANDMF (Choquette-Choo et al., 2023a), which showed how to perform privacy amplification with correlated noise. Other improvements include: extending DP-MF to adaptive optimizers (Ganesh et al., 2025), optimizing matrix factorization error (Kalinin et al., 2025), and leveraging special matrix structures to improve efficiency (Dvijotham et al., 2024; McMahan & Pillutla, 2025; Pillutla et al., 2025). To the best of our knowledge, we are the first to improve the quality of cross-iteration noise correlation using the model curvature information. **Using Public Data.** Pretraining a model on public data and then finetuning on private data using DP can achieve comparable utility as the non-private setting, across some ML tasks (Yu et al., 2022; Li et al., 2022a; De et al., 2022; Chua et al., 2024a; Kurakin et al., 2022). However, that comparable utility requires significant cost and a large-scale public dataset. For practitioners with a small amount of in-distribution public data can improve DP-SGD accuracy by using public data to pre-condition the private gradients by projecting those sensitive gradients onto a subspace estimated from the public data (Zhou et al., 2021; Yu et al., 2021). These techniques are orthogonal to, and can be used in conjunction with, our approach (which, by itself, does not need public data to be *labeled*). **Faster Convergence.** DP-SGD can be seen as a composition of individual DP mechanisms applied to each weight update, with the total privacy cost growing sublinearly of the total number of training iterations. Therefore, faster convergence for DP-SGD results

in lower privacy costs or, equivalently, lower noise scale under the same privacy budget. This line of work includes: using information from noisy history (Li et al., 2023), integrating second-order information (Ganesh et al., 2023), adaptively determining privacy parameters (Lee & Kifer, 2018; Bu et al., 2023a). Again, these improvements are orthogonal to our work. **Improving Efficiency.** DP-SGD also incurs significant computational overhead. Specifically, the requirement to clip per-sample gradients leads to extra memory usage and more training time. To address this, a line of work explores methods for performing gradient clipping without explicitly materializing per-sample gradients (Lee & Kifer, 2021; Li et al., 2022b; Bu et al., 2023b). Another source of inefficiency arises from privacy accounting: obtaining privacy amplification requires Poisson sampling, a strict data participation scheme that is difficult to parallelize. Recent work has therefore studied alternatives that preserve privacy amplification while avoiding Poisson sampling (Chua et al., 2024b; Ganesh, 2025; Feldman & Shenfeld, 2025; Balle et al., 2020).

## 2 MOTIVATION

Given a model $\Phi$ with learnable weights $\boldsymbol{w} \in \mathbb{R}^p$ and a loss function $\mathcal{L}$, let $\boldsymbol{w}_0, \boldsymbol{w}_1, \boldsymbol{w}_2, \ldots, \boldsymbol{w}_{\mathfrak{T}}$ represent the weights after each update of SGD. That is, $\boldsymbol{w}_0$ are the initial weights and $\boldsymbol{w}_{i+1} = \boldsymbol{w}_i - \eta \nabla \mathcal{L}(\boldsymbol{w}_i)$, where $\eta$ is the learning rate. To simplify the discussion for this section, we assume all gradients have norm smaller than some constant $\zeta$ (e.g., the gradients are clipped). For every iteration $i$, DP-SGD adds isotropic Gaussian noise $\boldsymbol{z}_i$, where $\boldsymbol{z}_i \sim N(\boldsymbol{0}, \boldsymbol{I}_p)$, to each (clipped) gradient, resulting in the weight sequence $\widetilde{\boldsymbol{w}}_0, \widetilde{\boldsymbol{w}}_1, \widetilde{\boldsymbol{w}}_2, \ldots, \widetilde{\boldsymbol{w}}_{\mathfrak{T}}$, where $\widetilde{\boldsymbol{w}}_0 = \boldsymbol{w}_0$ and $\widetilde{\boldsymbol{w}}_{i+1} = \widetilde{\boldsymbol{w}}_i - \eta(\nabla \mathcal{L}(\widetilde{\boldsymbol{w}}_i) + \boldsymbol{z}_i)$. The Gaussian noise vectors in different iterations are independent of each other: $\boldsymbol{z}_i \perp\!\!\!\perp \boldsymbol{z}_j$ when $j \neq i$.

The DP-MF line of work introduces correlations between noise added to different iterations. It searches for a mixing matrix $\boldsymbol{C} \in \mathbb{R}^{\mathfrak{T} \times \mathfrak{T}}$ and uses it to generate new noise vectors $\widetilde{\boldsymbol{z}}_0, \ldots \widetilde{\boldsymbol{z}}_{\mathfrak{T}-1}$ from the $\boldsymbol{z}_0, \ldots, \boldsymbol{z}_{\mathfrak{T}-1}$ as follows: $[\widetilde{\boldsymbol{z}}_0, \ldots \widetilde{\boldsymbol{z}}_{\mathfrak{T}-1}] = [\boldsymbol{z}_0, \boldsymbol{z}_2, \ldots, \boldsymbol{z}_{\mathfrak{T}-1}]\boldsymbol{C}^{-T}$. Thus, each $\widetilde{\boldsymbol{z}}_i$ is a linear combination of $\boldsymbol{z}_0, \boldsymbol{z}_2, \ldots, \boldsymbol{z}_{\mathfrak{T}-1}$. The update equation changes from $\widetilde{\boldsymbol{w}}_{i+1} = \widetilde{\boldsymbol{w}}_i - \eta(\nabla \mathcal{L}(\widetilde{\boldsymbol{w}}_i) + \boldsymbol{z}_i)$ to $\widetilde{\boldsymbol{w}}_{i+1} = \widetilde{\boldsymbol{w}}_i - \eta(\nabla \mathcal{L}(\widetilde{\boldsymbol{w}}_i) + \widetilde{\boldsymbol{z}}_i)$.

A proper choice of $\boldsymbol{C}$ can let DP-MF outperform DP-SGD at the same privacy cost (Denisov et al., 2022; Choquette-Choo et al., 2023a). In the current state-of-the-art methods (e.g., DP-BANDMF), $\boldsymbol{C}$ is obtained by solving a data-independent optimization problem (Denisov et al., 2022; Choquette-Choo et al., 2023a) to approximately minimize the sum of the squared errors of $\widetilde{\boldsymbol{w}}_1, \widetilde{\boldsymbol{w}}_2, \ldots, \widetilde{\boldsymbol{w}}_{\mathfrak{T}}$ (i.e., minimize $\sum_{i=0}^{\mathfrak{T}-1} ||\widetilde{\boldsymbol{z}}_i||_2^2$), subject to privacy constraints. That is, the error of the parameter estimate $\widetilde{\boldsymbol{w}}_i$ is modeled only through $||\widetilde{\boldsymbol{z}}_{i-1}||_2^2$, the amount of noise added in the $(i-1)^{\text{th}}$ iteration.

However, the difference between $\widetilde{\boldsymbol{w}}_i$ and $\boldsymbol{w}_i$ depends not just on the noise used in iteration $i-1$; it also depends on how prior noise had altered where the gradients are taken, i.e., $\nabla \mathcal{L}(\widetilde{\boldsymbol{w}}_{i-1})$ vs. $\nabla \mathcal{L}(\boldsymbol{w}_{i-1})$. To see this more clearly, consider the case of $i = 0$ and $i = 1$. Recall that $\boldsymbol{w}_0$ is the initial value of the weights and is the same for both private and non-private training:

$$\widetilde{\boldsymbol{w}}_1 - \boldsymbol{w}_1 = (\boldsymbol{w}_0 - \eta(\nabla \mathcal{L}(\boldsymbol{w}_0) + \widetilde{\boldsymbol{z}}_0)) - (\boldsymbol{w}_0 - \eta \nabla \mathcal{L}(\boldsymbol{w}_0)) = -\eta \widetilde{\boldsymbol{z}}_0$$
$$\widetilde{\boldsymbol{w}}_2 - \boldsymbol{w}_2 = (\widetilde{\boldsymbol{w}}_1 - \eta(\nabla \mathcal{L}(\widetilde{\boldsymbol{w}}_1) + \widetilde{\boldsymbol{z}}_1)) - (\boldsymbol{w}_1 - \eta \nabla \mathcal{L}(\boldsymbol{w}_1))$$
$$= (\widetilde{\boldsymbol{w}}_1 - \boldsymbol{w}_1) - \eta \widetilde{\boldsymbol{z}}_1 - \eta(\nabla \mathcal{L}(\widetilde{\boldsymbol{w}}_1) - \nabla \mathcal{L}(\boldsymbol{w}_1))$$
$$= -\eta \widetilde{\boldsymbol{z}}_0 - \eta \widetilde{\boldsymbol{z}}_1 - \eta(\nabla \mathcal{L}(\widetilde{\boldsymbol{w}}_1) - \nabla \mathcal{L}(\boldsymbol{w}_1))$$
$$\approx -\eta \widetilde{\boldsymbol{z}}_0 - \eta \widetilde{\boldsymbol{z}}_1 - \eta \nabla^2 \mathcal{L}(\boldsymbol{w}_1)(\widetilde{\boldsymbol{w}}_1 - \boldsymbol{w}_1)$$
$$= -\eta(\widetilde{\boldsymbol{z}}_0 + \widetilde{\boldsymbol{z}}_1) + \eta^2 \nabla^2 \mathcal{L}(\boldsymbol{w}_1)\widetilde{\boldsymbol{z}}_0.$$

While previous works on DP-MF optimize the first term ($-\eta(\widetilde{\boldsymbol{z}}_0 + \widetilde{\boldsymbol{z}}_1)$), they largely ignore the affect of the second term ($\eta^2 \nabla^2 \mathcal{L}(\boldsymbol{w}_1)\widetilde{\boldsymbol{z}}_0$), which is another source of error. The second term shows that the Hessian of the loss (or model curvature), along with the weight history, plays an important role in determining how the privacy-preserving update path $\boldsymbol{w}_0, \widetilde{\boldsymbol{w}}_1, \widetilde{\boldsymbol{w}}_2, \ldots$ diverges from the non-private path $\boldsymbol{w}_0, \boldsymbol{w}_1, \boldsymbol{w}_2, \ldots$. Our goal is to incorporate this curvature information into the optimization problem for the matrix $\boldsymbol{C}$ that determines the cross-iteration noise correlation. In Section 3, we use the quadratic case to derive an alternative objective function. Then, we discuss how to obtain approximate curvature information for realistic deep learning settings in Section 4.

## 3 A New Objective Function for the Mixing Matrix

Our proposed objective is based on insights obtained by examining gradient descent with correlated noise for the quadratic loss with known Hessian. In Section 4, we show how to approximately obtain curvature information can be incorporated into the DP-MF frameworks for deep learning, consisting of differentially private SGD with correlated (cross-iteration) noise and privacy amplification.

Let $\mathcal{L}(\boldsymbol{w}) = \frac{1}{2}(\boldsymbol{w} - \boldsymbol{d})^T \boldsymbol{H}(\boldsymbol{w} - \boldsymbol{d})$ be the loss function for some fixed vector $\boldsymbol{d}$ and positive semi-definite hessian $\boldsymbol{H} \in \mathbb{R}^{p \times p}$. Note that $\nabla \mathcal{L}(\boldsymbol{w}) = \boldsymbol{H}(\boldsymbol{w} - \boldsymbol{d})$ and $\nabla^2 \mathcal{L}(\boldsymbol{w}) = \boldsymbol{H}$. The noise-free update path, starting from initial weights $\boldsymbol{w}_0$ is $\boldsymbol{w}_0, \boldsymbol{w}_1, \ldots, \boldsymbol{w}_{\mathfrak{T}}$ where $\boldsymbol{w}_i = \boldsymbol{w}_{i-1} - \eta \nabla \mathcal{L}(\boldsymbol{w}_{i-1}) = \boldsymbol{w}_{i-1} - \eta \boldsymbol{H}(\boldsymbol{w}_{i-1} - \boldsymbol{d})$. Given i.i.d. vectors $\boldsymbol{z}_0, \ldots, \boldsymbol{z}_{\mathfrak{T}-1}$ with $\boldsymbol{z}_i \sim N(0, \boldsymbol{I}_p)$, the correlated noise vectors are $[\widetilde{\boldsymbol{z}}_0, \ldots, \widetilde{\boldsymbol{z}}_{\mathfrak{T}-1}] = [\boldsymbol{z}_0, \ldots, \boldsymbol{z}_{\mathfrak{T}-1}]\boldsymbol{C}^{-T}$. The correlated-noise update path, starting from initial weights $\widetilde{\boldsymbol{w}}_0 = \boldsymbol{w}_0$ is $\widetilde{\boldsymbol{w}}_0, \widetilde{\boldsymbol{w}}_1, \ldots, \widetilde{\boldsymbol{w}}_{\mathfrak{T}}$ where $\widetilde{\boldsymbol{w}}_i = \widetilde{\boldsymbol{w}}_{i-1} - \eta(\boldsymbol{H}(\widetilde{\boldsymbol{w}}_{i-1} - \boldsymbol{d}) + \widetilde{\boldsymbol{z}}_{i-1})$. The following theorem computes the difference in loss between the last noise-free iteration $\mathcal{L}(\boldsymbol{w}_{\mathfrak{T}})$ and the last correlated-noise iteration $\mathcal{L}(\widetilde{\boldsymbol{w}}_{\mathfrak{T}})$.

**Theorem 1.** *Under quadratic loss, define $\boldsymbol{X} = \boldsymbol{C}^T \boldsymbol{C}$. Let $\mu_0, \ldots, \mu_{p-1}$ be the eigenvalues (including multiplicity) of $\boldsymbol{H}$. Define the diagonal matrix $\boldsymbol{M} = Diag(\mu_0, \ldots, \mu_{p-1})$ and the matrix $\boldsymbol{V} \in \mathbb{R}^{p \times \mathfrak{T}}$ so that $\boldsymbol{V}[i, j] = (1 - \eta\mu_i)^{\mathfrak{T}-j-1}$ (indexing starts at 0). Then, the difference in expected loss between gradient descent with correlated noise and noise-free gradient descent is:*

$$E[\mathcal{L}(\widetilde{\boldsymbol{w}}_{\mathfrak{T}})] - E[\mathcal{L}(\boldsymbol{w}_{\mathfrak{T}})] = \frac{\eta^2}{2} \operatorname{Tr}(\boldsymbol{X}^{-1}(\boldsymbol{V}^T \boldsymbol{M} \boldsymbol{V})).$$

The proof can be found in Appendix B. Note that only the *eigenvalues* of the Hessian are important in expressing the loss difference. Challenges of estimating the eigenvalues of the Hessian and dealing with the fact that the Hessian is not constant in deep learning are handled in the following sections. Once these challenges are overcome, slotting Theorem 1 into the DP-BandMF framework is relatively straightforward, as we discuss next. Hence, let us temporarily assume that a single "representative" Hessian has been determined and its eigenvalues are computed.

Theorem 1 can slot directly into DP-BANDMF (Choquette-Choo et al., 2023a) that incorporates correlated noise with privacy amplification via batches. In the version of DP they use, two datasets $D$ and $D'$ are neighbors if one can be obtained from the other by blanking out 1 record.

**Definition 1** (Dwork et al. (2006a)). *Given privacy parameters $\epsilon > 0$ and $\delta \in [0, 1]$, a mechanism (randomized algorithm) $M$ satisfies $(\epsilon, \delta)$-differential privacy if for all pairs of neighboring datasets $D, D'$ and all subsets $S \subset range(M)$, $P(M(D) \in S) \leq e^{\epsilon}P(M(D') \in S) + \delta$.*

Given the privacy parameters $\epsilon, \delta$, and a desired iteration count $\mathfrak{T}$, one first adds privacy constraints that `diag`$(\boldsymbol{X}) = \boldsymbol{1}$ (i.e., each diagonal entry of $\boldsymbol{X} \equiv \boldsymbol{C}^T \boldsymbol{C}$ is 1) and $\boldsymbol{X}$ is positive definite and banded (which enables privacy amplification). In other words, $\boldsymbol{X}_{i,j} = 0$ when $|i - j| \geq b$ (the band size $b$ is a tunable hyperparameter). Using the notation of Theorem 1, replacing the objective function of DP-BANDMF gives us the following optimization problem:

$$
\begin{aligned}
\boldsymbol{X} \quad &\leftarrow \arg\min \operatorname{Tr}((\boldsymbol{V}^T \boldsymbol{M} \boldsymbol{V})\boldsymbol{X}^{-1}) \\
\text{subject to} \quad &\texttt{diag}(\boldsymbol{X}) = \boldsymbol{1} \qquad\qquad \text{(privacy constraint)} \\
&\boldsymbol{X} \succ 0 \\
&\boldsymbol{X}_{i,j} = 0, \forall |i - j| \geq b \qquad (\boldsymbol{X} \text{ is } b\text{-banded})
\end{aligned}
\tag{P1}
$$

As we have only changed the objective function, as in the original DP-BANDMF (Choquette-Choo et al., 2023a), this convex problem that can be solved by optimizers like L-BFGS (Nocedal, 1980) with projections onto the feasible region to enforce `diag`$(\boldsymbol{X}) = \boldsymbol{1}$, and starting with an initial guess for $\boldsymbol{X}$ that is symmetric and positive definite. Following DP-BANDMF, once an $\boldsymbol{X} \equiv \boldsymbol{C}^T \boldsymbol{C}$ is obtained, one can use the Cholesky decomposition to recover $\boldsymbol{C}$ as a lower triangular matrix. The rest of the DP-BANDMF framework (see Choquette-Choo et al. (2023a) for full details) works as follows. **Batch Creation:** given a band hyperparameter $b$ and batch size $|B|$, the dataset $D$ is initially divided into $b$ equal sized partitions $D_1, \ldots, D_b$. During model training, the batch $B_t$ for iteration $t$ is obtained by (1) selecting partition $D_\ell$, where $\ell = 1 + (t \mod b)$ then (2) setting the batch $B_t$ to be a random sample from $D_\ell$ of size $|B|$. **Noise Correlation:** using $\boldsymbol{C}$ to obtain correlated noise $\widetilde{\boldsymbol{z}}_0, \ldots, \widetilde{\boldsymbol{z}}_{\mathfrak{T}-1}$ in an online manner is essentially Gaussian elimination (see Choquette-Choo

et al. (2023a) for full details). **Parameter Updates:** as with normal DP-SGD, the gradient of each example within a batch is clipped so that its norm is at most some hyperparameter $\zeta$. The clipped gradients in a batch are averaged and noise $\zeta\sigma\widetilde{z}_i$ is added to it ($\sigma$ is the noise multiplier that decides the differential privacy parameters). The noisy gradient is fed into an optimizer like Adam (Kingma & Ba, 2017). **Privacy Accounting:** can be performed using existing libraries like Opacus (Yousef-pour et al., 2021). Specifically, given a target $\epsilon$ and $\delta$, the Opacus privacy accountant can give the proper noise multiplier $\sigma$ to use when one sets its sampling probability parameter $q = |B|/\lfloor\frac{|D|}{b}\rfloor$ and its "number of compositions" to be $\lceil\frac{\mathfrak{T}|B|^2}{q|D|}\rceil$.

# 4 PUTTING THE PIECES TOGETHER FOR REAL DATA AND MODELS

Extending the result from quadratic loss (Section 3) to real-world deep neural network (DNN) training poses four challenges, **C1–4**: (**C1**) The Hessian and its eigenvalues are unknown, but estimating them from the (private) training dataset would incur additional privacy leakage. (**C2**) Unlike the quadratic case, the Hessian and its eigenvalues constantly change. This deviates from the constant Hessian setting of Theorem 1. (**C3**) The eigenvalues of the Hessian can become negative for non-convex models/loss, which makes Problem P1 non-convex and hard to solve. (**C4**) Finally, estimating all Hessian eigenvalues for a large model becomes computationally infeasible.

In this section, we present solutions (**S1–4**) to these challenges (**C1–4**), explaining the motivation and providing empirical support. The guiding theme is that *some* curvature information is often better than none (i.e., the standard DP-MF objective). The solutions are simplified by the fact that the workload matrix $V^T M V$ used in the objective function (Problem P1) only depends on *eigenvalues* of the Hessian, instead of the full Hessian. The solution overview is: (**S1**) To estimate the eigenvalues without leaking private data, we use unlabeled public data (which is much easier to collect than labeled data). We empirically observe that the eigenvalue distribution from public data reasonably matches that of the target data, at least in the case of images (Section 4.1). (**S2**) To cope with the changing Hessian of non-quadratic loss, we pretrain the model with an **unlabeled** public dataset (by creating random label assignments) and estimate eigenvalues once when pretraining ends. This becomes a representative exemplar of typical Hessian eigenvalues in the subsequent training on the target dataset and is plugged into the objective function. This is supported by observations that the eigenvalues do not change significantly after pretraining (Section 4.2). (**S3**) We handle the negative eigenvalues of non-convex loss through simply zeroing them out (Section 4.3). This is motivated by the fact that if model training reaches a stable local minimum, the Hessian would have very few, if any, large negative eigenvalues (i.e., large negative eigenvalues indicate an unstable saddle point, and it is unlikely that convergence would happen there). (**S4**) For large models, eigenspectrum computation is infeasible. We introduce a new method to approximate the positive eigenspectrum by combining top-$k$ eigenvalue estimation with estimation of the total number of positive eigenvalues (Section 4.4).

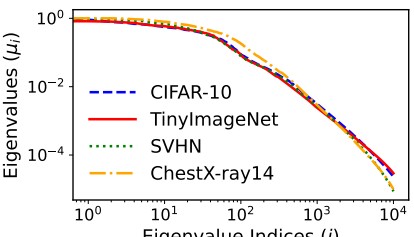

(a) CNN with GroupNorm

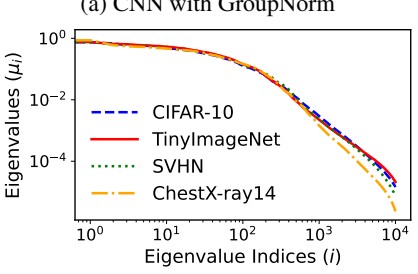

(b) CNN with Maxpool

Figure 2: Largest 10,000 eigenvalues of four datasets estimated on two CNNs. Setup details are in Appendix D.1.

## 4.1 S1: ESTIMATE THE HESSIAN WITH A PUBLIC DATASET

The objective function (Theorem 1) requires knowing $\mu_i$, the eigenvalues of the Hessian $H$. We want to avoid using private data for Hessian estimation as this would incur additional privacy loss. Instead, we use *unlabeled public data*. Specifically, we use a self-supervised learning technique, SimCLR (Chen et al., 2020), for pretraining, which does not require the public data to be labeled. We reuse the same public data for both pretraining and the eigenvalue estimation (Section 4.1). Us-

ing the public data instead of target data for Hessian estimation may seem counterintuitive, but is supported by prior work. For example, Sagun et al. (2016) empirically observed that a large portion of the Hessian spectrum, except for a spike of a few large positive values, is governed by the model architecture, not data. Papyan (2019) observed that the bulk of the Hessian spectrum is originated from the model curvature, rather than the (data-dependent) loss curvature. The importance of architecture is also emphasized by Ulyanov et al. (2018), who noted that computer vision models, even with randomly initialized weights, provide acceptable results for many tasks. A common hypothesis in the literature is that neural networks must implicitly perform a variety of common image tasks, like edge detection, texture-based segmentation, etc., and useful architectures result in similarities in the eigenspectrum on different datasets.

We provide further empirical results in Figure 2, which shows the distribution of the Hessian eigenvalues estimated with four different datasets on two small pretrained CNNs. The four datasets were CIFAR-10 (Krizhevsky et al., 2009), TinyImageNet (Le & Yang, 2015), SVHN (Netzer et al., 2011), and ChestX-ray14 (Wang et al., 2017). Details on the architectures can be found in Appendix D.1. The figures show that, for both CNNs, the eigenvalue distributions closely overlap if similar datasets are used (CIFAR-10 and TinyImageNet), which is expected. But, even for datasets that are fairly different in nature, like SVHN (street view house numbers) and ChestX-ray14 (X-ray images), the eigenspectra deviate only slightly from those of CIFAR-10 and TinyImageNet. Furthermore, we will later show in our evaluation (Section 5.2) that using these not-so-close datasets to estimate eigenvalues and create noise correlations can still improve the accuracy of privacy-preserving model training (i.e., some curvature information is better than none).

## 4.2 S2: Use a representative Hessian even though true Hessian is not constant

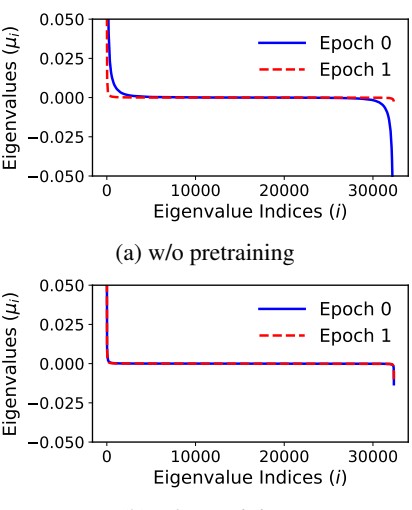

(a) w/o pretraining

(b) w/ pretraining

Figure 3: Change of the eigenvalues during training, (a) without and (b) with pretraining. Only values between -0.05 and 0.05 are zoomed in. Experiment details are in Appendix D.1.

The derivation of Theorem 1 relies on the fact that the Hessian and its eigenvalue are constant throughout training, which is not the case for DNN training with a non-quadratic loss. However, we empirically observed that the eigenvalues of the Hessian do not change significantly after the model is properly pretrained. Figure 3 shows that, *without* pretraining (Figure 3a), the eigenvalue distribution at the start of training epoch 0 and epoch 1 changes drastically. However, a model pretrained on unlabeled data (Figure 3b) does not exhibit such behavior, with the eigenvalues from epoch 0 and epoch 1 being nearly identical, indicating that the eigenspectrum is likely to be very stable from this point on. Hence, after the pre-training approach from Solution **S1**, the resulting eigenvalues from public data are expected to be good representatives of the true eigenvalues of Hessians along the privacy-preserving learning trajectory.

Comparing Figure 3a and Figure 3b also shows that the negative eigenvalues becomes closer to zero after pre-training, which helps in handling negative eigenvalues in Section 4.3. Recent work (Ghorbani et al., 2019; Gur-Ari et al., 2018) similarly observed during CNN training that the Hessian spectrum stabilizes after a few iterations, and large negative eigenvalues quickly disappear. The observations align with our own empirical findings.

## 4.3 S3: Truncating Negative Eigenvalues for Nonconvex Losses

When the loss in non-convex, the eigenvalues of the Hessian can become negative, and the workload matrix $V^T M V$ in Problem P1 is not positive semidefinite (recall that $M$ is a diagonal matrix whose elements are the eigenvalues of the Hessian). This makes the overall problem non-convex and can make it infeasible to solve. Fortunately, our earlier observations from Section 4.2 (Figure 3b) have already showed that the negative eigenvalues become close to zero after pretraining the model, unlike

a randomly initialized model (Figure 3a). Consistently, Ghorbani et al. (2019) reported that large negative eigenvalues vanish within a few training iterations in larger CNNs. These results indicate that the Hessian at $w_t$ (after training progresses or pretraining) becomes nearly positive semidefinite, implying that optimization typically (and intuitively) converges in a locally convex neighborhood of the loss surface rather than around a high-curvature saddle point. Leveraging this observation, we simply *replace the negative eigenvalues with zero*. Doing so makes Problem P1 convex and solvable.

## 4.4 S4: Approximating the Eigenvalues Through Curve-Fitting

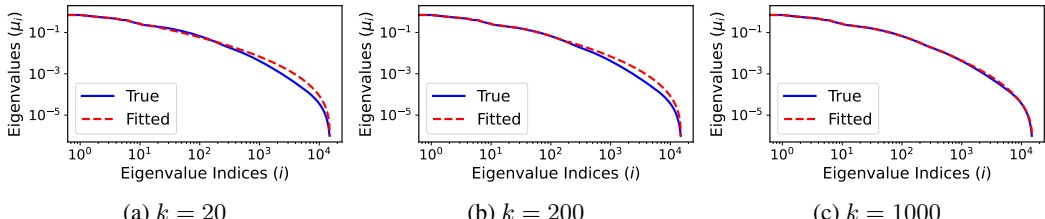

(a) $k = 20$       (b) $k = 200$       (c) $k = 1000$

Figure 4: The true (positive) eigenvalues (in descending order) of a small CNN ($\sim$30,000 parameters) on CIFAR-10 validation set and our approximated curve, using the top $k = 20, 200, 1000$ eigenvalues for fitting. We used $p_+$=12,000 and $\mu_{p_+} = 10^{-6}$. Details of the setup can be found in Appendix D.1.

The number of eigenvalues of a Hessian equals to the number of parameters in a model, and calculating the full eigenspectrum is too expensive for large models. While many of eigenvalues are near zero (Figure 3b), we have observed that the number of relatively large eigenvalues are still in the order of tens or hundreds of thousands (Figure 2). Calculating this many eigenvalues was not possible with our computing resources (we could only calculate in the order of hundreds to thousands), so we introduce a new compute-efficient approximation. Design of this approximation is an orthogonal contribution to the rest of our paper and can be applied to other applications where the eigenvalues need to be estimated.

The approximation happens in a two-step process. First, we estimate the $k$ largest positive eigenvalues up to a point allowed by the compute budget. With our setup, we were able to compute the $k$=300–2000 largest eigenvalues directly, using the Lanczos (Lanczos, 1950) algorithm. Typically, the magnitudes of the positive eigenvalues drop off quickly, but we observed that the ones after the top-$k$ still represent a substantial long tail. Simply setting them to 0 would be incorrect, because if an estimated Hessian $H_{est}$ has rank $k$, it implies that the gradients can only change inside a rank-$k$ subspace. If the true Hessian $H_{true}$ has a longer tail in its eigenspectrum, the actual gradients can have large total deviations outside the rank $k$ subspace.

Instead, we extrapolate the non-negligible tail by fitting a smooth, simple curve. Inspired by the empirical shape of the eigenvalues of *small* models for which we could compute the eigenspectrum, (Figures 2, 3b), the eigenspectrum looks like following a power-law on the log-log scale. So, we fit a curve of the following form:

$$\log \mu_i = C \cdot (-\log i - \log p_+)^\alpha + \log \mu_{p_+}, \tag{2}$$

where $\mu_i$ is the $i^{\text{th}}$ eigenvalue, sorted in non-increasing order; $p_+$ is the position (index) where the long tail ends — as it is not known, it can be treated as a deep learning hyperparameter (it typically ranged from 100k–200k in our experiments); and $\mu_{p_+}$ is the magnitude of the eigenvalue at position $p_+$ — this could either be treated as a deep learning hyperparameter or, since $\mu_{p_+}$ should be barely above zero, one could choose some small value ($10^{-5}$–$10^{-6}$); $C$ and $\alpha$ are the coefficients to be fitted.

For small models, whose eigenspectrum is feasible to compute, Figure 4 shows that the fitted curve is a reasonable approximation of the ground truth. As expected, the approximation error decreases as $k$ increases. When $k$=1000, the fitted curve is visually nearly identical to the true distribution. Table 3 in Appendix C additionally shows that using the approximated eigenvalues to solve Problem P1 effectively reduces the objective. For large models whose eigenspectrum is infeasible to

compute, we cannot directly compare to the ground truth. Hence, we rely on an indirect assessment, evaluating whether the approximated eigenvalues could produce noise correlations that improve privacy-preserving model training (experiments in Section 5).

## 5 EXPERIMENTS

We compare our proposed NoiseCurve to DP-BANDMF and DP-SGD on the same learning tasks under various privacy budgets $\epsilon$. We use CIFAR-10 as a standard vision benchmark and ChestX-ray14 as a medical X-ray dataset, chosen for its relevance to privacy-sensitive applications. Our models include several CNN-based architectures for CIFAR10 — a pretrained ResNet, a small CNN, and a modified large-scale VGG—as well as a Transformer-based Vision Transformer (ViT) for ChestX-ray14. We experiment with both full-parameter training and parameter-efficient fine-tuning. We use TinyImageNet for pretraining and Hessian estimation. Notably, we split 10% of data samples from training sets as validation sets and **tune hyperparameters on validation sets** (no tuning on the test sets). Further details on our experimental setup (datasets, models, hyperparameters, etc.) appear in Appendix D.1.

Since our new mixing matrix $C$ is optimal under quadratic loss, our experiments explore step-by-step the gap between analytically optimal and practical setups. We begin with a convex loss setup where we privately finetune the last layer of a pretrained ResNet, then proceed to fully finetuning a nonconvex but small-scale CNN, where the full Hessian is feasible to compute on our computers (Section 5.1). Finally, we arrive at more common non-convex loss models (Section 5.2), where full Hessian computation is infeasible but its eigenvalues can be approximated by the technique introduced in Section 4.4. Results show that our new noise correlation achieves **1–4% higher accuracy** than DP-SGD and DP-BANDMF across all privacy budgets we test.

We first note that our results for DP-BANDMF differ from those reported by Choquette-Choo et al. (2023a). We could only match those results if we allowed DP-BANDMF to use the test set for hyperparameter tuning. As this is also a common practice in the differentially private literature, Appendix D.2 shows results when all methods are tuned on the test set (NoiseCurve still consistently outperforms competitors). Another minor experiment (Appendix D.3) shows that eigenvalues estimated from TinyImageNet are nearly as good as those estimated from validation data.

### 5.1 RESULTS FOR CONVEX AND SMALL NONCONVEX MODELS

We compare our method with DP-SGD and DP-BANDMF on a canonical ML task — classifying CIFAR-10 with (1) finetuning the last layer of an ImageNet-pretrained ResNet152, where the loss is convex, (2) fully finetuning a small CNN, whose loss is nonconvex, under a wide range of privacy budget $\epsilon$. For both settings, Hessian matrices are computable. The evaluation results are in Table 1.

Table 1: Test mean accuracy and standard deviation (over 3 runs) on CIFAR-10 for $\delta = 10^{-5}$ and varying $\epsilon$ on (a) finetuning the last layer of ResNet152 (convex case), and (b) a small CNN (non-convex case). Both were pretrained with ImageNet, and full Hessian matrix was computable.

| (a) Last layer finetuning of ResNet152 | | | | | (b) Full finetuning of a small CNN | | | | |
|---|---|---|---|---|---|---|---|---|---|
| Non-private | $\epsilon$ | DP-SGD | DP-BANDMF | NoiseCurve | Non-private | $\epsilon$ | DP-SGD | DP-BANDMF | NoiseCurve |
| | 1 | $75.89 \pm 0.01$ | $75.94 \pm 0.04$ | $\mathbf{76.5 \pm 0.2}$ | | 1 | $48.65 \pm 0.32$ | $49.18 \pm 0.31$ | $\mathbf{49.8 \pm 0.24}$ |
| 82.01 | 2 | $77.15 \pm 0.18$ | $77.3 \pm 0.08$ | $\mathbf{78.29 \pm 0.05}$ | 72.99 | 2 | $51.81 \pm 0.33$ | $52.14 \pm 0.79$ | $\mathbf{54.13 \pm 0.46}$ |
| | 5 | $78.27 \pm 0.21$ | $78.33 \pm 0.06$ | $\mathbf{80.11 \pm 0.32}$ | | 5 | $54.6 \pm 0.7$ | $55.42 \pm 0.68$ | $\mathbf{58.07 \pm 0.17}$ |

Across both learning setups (convex and non-convex losses) and across the full range of $\epsilon$ we tested, our method consistently outperformed DP-SGD and DP-BANDMF. Notably, we observed a non-negligible improvement even in highly private regimes ($\epsilon$=1–2), where DP-BANDMF could not achieve a notable improvement over DP-SGD. The original authors of DP-BANDMF (Choquette-Choo et al., 2023a) also observed that their method could not do much better than DP-SGD under a small $\epsilon$, reporting that the best band size was 1 (which makes $C = I$, where DP-BANDMF coincides with DP-SGD). Our method, NoiseCurve, showed notable improvements even under a small $\epsilon$, finding a noise correlation matrix $C$ that is better than $I$. The result highlights the ability of our new objective, which better leverages the information from the network architecture.

## 5.2 RESULTS FOR LARGE NONCONVEX MODELS AND DIFFERENT DATASETS

For large models where the full Hessian is expensive or infeasible to compute, we incorporate the approximation techniques introduced in Section 4.4 to enable NoiseCurve for differentially private training large models. We evaluated our method on two setups: (1) full training of VGG, and (2) finetuning ViT with LoRA. We simultaneously evaluated the performance of our approach when the public data does not closely align with the target (private) data on the second setup, by using TinyImageNet for public data and ChestX-ray14, which contains a conceptually different classes of (medical) imagery, as the target data. The evaluation results are in Figure 1 and Table 2.

Table 2: Test accuracy comparison on the ChestX-ray14 dataset (ViT, LoRA finetuning) for $\delta = 10^{-5}$ as $\epsilon$ varies. We report the mean AUC for accuracy. We omit the result for $\epsilon$=1–2 as the accuracy was too low for all methods (50% AUC is random guessing).

| Non-private | $\epsilon$ | DP-SGD | DP-BANDMF | NoiseCurve |
|---|---|---|---|---|
| 73.63 | 5 | 58.39 | 59.71 | **62.67** |
| | 8 | 59.66 | 61.99 | **64.28** |

Results indicate that the accuracy improves even with the approximated eigenvalues, and Noise-Curve consistently outperforms both DP-SGD and DP-BANDMF. Remarkably, our experiments on the ChestX-ray14 dataset (Table 2) demonstrate that even under a significant domain shift (Tiny-ImageNet vs. ChestX-ray14), eigenvalues remain informative and contribute to a better $C$. We additionally tried NoiseCurve with using the validation set of CIFAR-10 and ChestX-ray14 directly for eigenvalue approximation, which represents cases where a public in-distribution dataset that is highly similar with the target dataset is available. While the accuracy improved with the better public dataset as expected, the accuracy improvement was only around 0.1–0.14% for CIFAR-10 and was a moderate 1.47–1.62% for ChestX-ray14. The result implies that a close dataset like TinyImageNet for CIFAR-10 is good enough as a public dataset. TinyImageNet also serves as a reasonable public dataset for ChestX-ray14, although the accuracy could further improve with an even similar public dataset. Details of these results are in Appendix D.3.

## 6 DISCUSSION AND LIMITATIONS

We propose NoiseCurve, which improves the quality of cross-iteration noise correlation through model curvature. Although NoiseCurve achieves non-trivial accuracy improvements over DP-SGD and DP-BANDMF baselines, we recognize a few limitations of this work that require further study.

**Public Data Dependence**  The effect of public-data choice on the quality of the learned noise correlation remains insufficiently understood. In Section 4.1, we report small-CNN evidence that (i) Hessian spectra computed on different vision datasets "look similar", and (ii) using TinyImageNet as public data improves accuracy for chest X-ray. A plausible explanation is that the bulk of the Hessian spectrum is induced primarily by model curvature rather than data-dependent loss curvature; consequently, for a fixed architecture, loss function, and suitably chosen parameter point, Hessian spectra across datasets can appear qualitatively similar. However, we do not yet have a formal analysis to support it, and it is unclear whether the observation holds beyond these specific settings (e.g., language or structured/tabular domains). Further theoretical and cross-domain empirical work is needed to characterize when public data choice matters for noise-correlation quality.

**Eigenvalue Approximation**  The effectiveness of the Hessian eigenvalue approximation technique introduced in Section 4.4 remains insufficiently explored. As it is orthogonal to the main contributions of this work, our validation has so far been limited to (i) qualitative comparison with ground-truth spectra on small CNNs, and (ii) indirect evidence from training accuracy when incorporated into our method. We believe, however, that this technique has broader potential in other areas that rely on Hessian spectral information. To support such applications, its accuracy on larger-scale models must be systematically evaluated. Moreover, its stability with respect to the endpoint eigenvalue, $(p_+, \mu_{p_+})$, requires deeper theoretical and empirical understanding.

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

## A    THE USE OF LARGE LANGUAGE MODELS (LLMs)

We use LLM-based software Grammarly to polish writing. Our three-step process involves writing the full text independently, submitting it to Grammarly for feedback, and then revising the text based on the suggestions for improved word choice and sentence structure.

## B    PROOFS

**Theorem 1.** *Under quadratic loss, define* $\boldsymbol{X} = \boldsymbol{C}^T\boldsymbol{C}$. *Let* $\mu_0, \ldots, \mu_{p-1}$ *be the eigenvalues (including multiplicity) of* $\boldsymbol{H}$. *Define the diagonal matrix* $\boldsymbol{M} = Diag(\mu_0, \ldots, \mu_{p-1})$ *and the matrix* $\boldsymbol{V} \in \mathbb{R}^{p \times \mathfrak{T}}$ *so that* $\boldsymbol{V}[i,j] = (1 - \eta\mu_i)^{\mathfrak{T}-j-1}$ *(indexing starts at 0). Then, the difference in expected loss between gradient descent with correlated noise and noise-free gradient descent is:*

$$E[\mathcal{L}(\widetilde{\boldsymbol{w}}_{\mathfrak{T}})] - E[\mathcal{L}(\boldsymbol{w}_{\mathfrak{T}})] = \frac{\eta^2}{2}\operatorname{Tr}(\boldsymbol{X}^{-1}(\boldsymbol{V}^T\boldsymbol{M}\boldsymbol{V})).$$

*Proof of Theorem 1.* The first step is to obtain closed-form expressions for $\widetilde{\boldsymbol{w}}_i - \boldsymbol{w}_i$, for each $i$, in terms of the noise vectors, Hessian, and initialization. We observe that

$$\widetilde{\boldsymbol{w}}_1 - \boldsymbol{w}_1 = (\boldsymbol{w}_0 - \eta(\boldsymbol{H}(\boldsymbol{w}_0 - \boldsymbol{d}) + \widetilde{\boldsymbol{z}}_0)) - (\boldsymbol{w}_0 - \eta\boldsymbol{H}(\boldsymbol{w}_0 - \boldsymbol{d})) = -\eta\widetilde{\boldsymbol{z}}_0$$

$$\widetilde{\boldsymbol{w}}_2 - \boldsymbol{w}_2 = (\widetilde{\boldsymbol{w}}_1 - \eta(\boldsymbol{H}(\widetilde{\boldsymbol{w}}_1 - \boldsymbol{d}) + \widetilde{\boldsymbol{z}}_1)) - (\boldsymbol{w}_1 - \eta\boldsymbol{H}(\boldsymbol{w}_1 - \boldsymbol{d}))$$

$$= (\widetilde{\boldsymbol{w}}_1 - \boldsymbol{w}_1) - \eta\boldsymbol{H}(\widetilde{\boldsymbol{w}}_1 - \boldsymbol{w}_1) - \eta\widetilde{\boldsymbol{z}}_1$$

$$= (\boldsymbol{I} - \eta\boldsymbol{H})(\widetilde{\boldsymbol{w}}_1 - \boldsymbol{w}_1) - \eta\widetilde{\boldsymbol{z}}_1 = -\eta(\boldsymbol{I} - \eta\boldsymbol{H})\widetilde{\boldsymbol{z}}_0 - \eta\widetilde{\boldsymbol{z}}_1$$

$$\widetilde{\boldsymbol{w}}_3 - \boldsymbol{w}_3 = (\boldsymbol{I} - \eta\boldsymbol{H})(\widetilde{\boldsymbol{w}}_2 - \boldsymbol{w}_2) - \eta\widetilde{\boldsymbol{z}}_2 = -\eta(\boldsymbol{I} - \eta\boldsymbol{H})^2\widetilde{\boldsymbol{z}}_0 - \eta(\boldsymbol{I} - \eta\boldsymbol{H})\widetilde{\boldsymbol{z}}_1 - \eta\widetilde{\boldsymbol{z}}_2$$

$$\widetilde{\boldsymbol{w}}_i - \boldsymbol{w}_i = -\eta\sum_{j=0}^{i-1}(\boldsymbol{I} - \eta\boldsymbol{H})^j\widetilde{\boldsymbol{z}}_{i-j-1} \tag{3}$$

Then we calculate the difference in losses for the final noisy and non-noisy parameters:

$$\mathcal{L}(\widetilde{\boldsymbol{w}}_{\mathfrak{T}}) - \mathcal{L}(\boldsymbol{w}_{\mathfrak{T}}) = \frac{1}{2}(\widetilde{\boldsymbol{w}}_{\mathfrak{T}} - \boldsymbol{d})^T\boldsymbol{H}(\widetilde{\boldsymbol{w}}_{\mathfrak{T}} - \boldsymbol{d}) - \frac{1}{2}(\boldsymbol{w}_{\mathfrak{T}} - \boldsymbol{d})^T\boldsymbol{H}(\boldsymbol{w}_{\mathfrak{T}} - \boldsymbol{d})$$

$$= \frac{1}{2}\left(\widetilde{\boldsymbol{w}}_{\mathfrak{T}} - \boldsymbol{w}_{\mathfrak{T}} + (\boldsymbol{w}_{\mathfrak{T}} - \boldsymbol{d})\right)^T\boldsymbol{H}\left(\widetilde{\boldsymbol{w}}_{\mathfrak{T}} - \boldsymbol{w}_{\mathfrak{T}} + (\boldsymbol{w}_{\mathfrak{T}} - \boldsymbol{d})\right) - \frac{1}{2}(\boldsymbol{w}_{\mathfrak{T}} - \boldsymbol{d})^T\boldsymbol{H}(\boldsymbol{w}_{\mathfrak{T}} - \boldsymbol{d})$$

$$= \frac{1}{2}(\widetilde{\boldsymbol{w}}_{\mathfrak{T}} - \boldsymbol{w}_{\mathfrak{T}})^T\boldsymbol{H}(\widetilde{\boldsymbol{w}}_{\mathfrak{T}} - \boldsymbol{w}_{\mathfrak{T}}) + (\widetilde{\boldsymbol{w}}_{\mathfrak{T}} - \boldsymbol{w}_{\mathfrak{T}})^T\boldsymbol{H}(\boldsymbol{w}_{\mathfrak{T}} - \boldsymbol{d})$$

$$E[\mathcal{L}(\widetilde{\boldsymbol{w}}_{\mathfrak{T}}) - \mathcal{L}(\boldsymbol{w}_{\mathfrak{T}})] = E\left[\frac{1}{2}(\widetilde{\boldsymbol{w}}_{\mathfrak{T}} - \boldsymbol{w}_{\mathfrak{T}})^T\boldsymbol{H}(\widetilde{\boldsymbol{w}}_{\mathfrak{T}} - \boldsymbol{w}_{\mathfrak{T}})\right] + E\left[\left(-\eta\sum_{j=0}^{\mathfrak{T}-1}(\boldsymbol{I} - \eta\boldsymbol{H})^j\widetilde{\boldsymbol{z}}_{\mathfrak{T}-j-1}\right)^T\boldsymbol{H}(\boldsymbol{w}_{\mathfrak{T}} - \boldsymbol{d})\right]$$

$$= E\left[\frac{1}{2}\left(-\eta\sum_{j=0}^{\mathfrak{T}-1}(\boldsymbol{I} - \eta\boldsymbol{H})^j\widetilde{\boldsymbol{z}}_{\mathfrak{T}-j-1}\right)^T\boldsymbol{H}\left(-\eta\sum_{j=0}^{\mathfrak{T}-1}(\boldsymbol{I} - \eta\boldsymbol{H})^j\widetilde{\boldsymbol{z}}_{\mathfrak{T}-j-1}\right)\right]$$

$$= E\left[\frac{1}{2}\left(-\eta\sum_{j=0}^{\mathfrak{T}-1}(\boldsymbol{I} - \eta\boldsymbol{H})^j(\sum_i C^{-1}[\mathfrak{T} - j - 1, i]\boldsymbol{z}_i)\right)^T\boldsymbol{H}\left(-\eta\sum_{j=0}^{\mathfrak{T}-1}(\boldsymbol{I} - \eta\boldsymbol{H})^j(\sum_i C^{-1}[\mathfrak{T} - j - 1, i]\boldsymbol{z}_i)\right)\right]$$

$$= \frac{\eta^2}{2}\sum_{j=0}^{\mathfrak{T}-1}\sum_{\ell=0}^{\mathfrak{T}-1}\sum_i E\left[\left((\boldsymbol{I} - \eta\boldsymbol{H})^j C^{-1}[\mathfrak{T} - j - 1, i]\boldsymbol{z}_i\right)^T\boldsymbol{H}\left((\boldsymbol{I} - \eta\boldsymbol{H})^\ell C^{-1}[\mathfrak{T} - \ell - 1, i]\boldsymbol{z}_i\right)\right]$$

$$= \frac{\eta^2}{2}\sum_{j=0}^{\mathfrak{T}-1}\sum_{\ell=0}^{\mathfrak{T}-1}\sum_i E\left[\left(C^{-1}[\mathfrak{T} - j - 1, i]\boldsymbol{z}_i\right)^T\boldsymbol{H}(\boldsymbol{I} - \eta\boldsymbol{H})^{j+\ell}\left(C^{-1}[\mathfrak{T} - \ell - 1, i]\boldsymbol{z}_i\right)\right]$$

$$= \frac{\eta^2}{2}\sum_{j=0}^{\mathfrak{T}-1}\sum_{\ell=0}^{\mathfrak{T}-1}\sum_i C^{-1}[\mathfrak{T} - j - 1, i]C^{-1}[\mathfrak{T} - \ell - 1, i]\operatorname{trace}(\boldsymbol{H}(\boldsymbol{I} - \eta\boldsymbol{H})^{j+\ell})$$

Now define $\boldsymbol{X} = \boldsymbol{C}^T \boldsymbol{C}$

$$= \frac{\eta^2}{2} \sum_{j=0}^{\mathfrak{T}-1} \sum_{\ell=0}^{\mathfrak{T}-1} \boldsymbol{X}^{-1}[\mathfrak{T}-j-1, \mathfrak{T}-\ell-1] \mathrm{trace}(\boldsymbol{H}(\boldsymbol{I}-\eta\boldsymbol{H})^{j+\ell})$$

$$= \frac{\eta^2}{2} \sum_{j=0}^{\mathfrak{T}-1} \sum_{\ell=0}^{\mathfrak{T}-1} \boldsymbol{X}^{-1}[j, \ell] \mathrm{trace}(\boldsymbol{H}(\boldsymbol{I}-\eta\boldsymbol{H})^{2\mathfrak{T}-j-\ell-2})$$

Let $\mu_0, \ldots, \mu_{p-1}$ be the eigenvalues (including multiplicity) of $\boldsymbol{H}$ sorted in non-decreasing order

$$= \frac{\eta^2}{2} \sum_{j=0}^{\mathfrak{T}-1} \sum_{\ell=0}^{\mathfrak{T}-1} \boldsymbol{X}^{-1}[j, \ell] \sum_{i=0}^{p-1} \mu_i (1-\eta\mu_i)^{\mathfrak{T}-j-1}(1-\eta\mu_i)^{\mathfrak{T}-\ell-1}$$

Define the diagonal matrix $\boldsymbol{M} = \mathrm{Diag}(\mu_0, \ldots, \mu_{p-1})$ and the matrix $\boldsymbol{V} \in \mathbb{R}^{p \times \mathfrak{T}}$ so that $\boldsymbol{V}[i, j] = (1-\eta\mu_i)^{\mathfrak{T}-j-1}$

$$= \frac{\eta^2}{2} \sum_{j=0}^{\mathfrak{T}-1} \sum_{\ell=0}^{\mathfrak{T}-1} \boldsymbol{X}^{-1}[j, \ell](\boldsymbol{V}^T \boldsymbol{M} \boldsymbol{V})[j, \ell] = \frac{\eta^2}{2} \mathrm{trace}(\boldsymbol{X}^{-1}(\boldsymbol{V}^T \boldsymbol{M} \boldsymbol{V}))$$

$\square$

## C  Spectrum Curve Fitting

In this section, we present additional quantitative evidence to support the efficacy of curve-fitting stargetig proposed in Section 4.4. We measure it under reduction in objective — the objective difference when substituting the optimal solution of Problem P1 to an approximated one:

$$\mathrm{Tr}((\boldsymbol{V}^\top \boldsymbol{M} \boldsymbol{V}) \boldsymbol{X}_{\mathrm{approx}}^{-1}) - \mathrm{Tr}((\boldsymbol{V}^\top \boldsymbol{M} \boldsymbol{V}) \boldsymbol{X}_*^{-1})$$

where $\boldsymbol{X}_{\mathrm{approx}}$ is given by using $(\boldsymbol{V}^\top \boldsymbol{M} \boldsymbol{V})_{\mathrm{approx}}$ built on top of eigenvalues other than the ground truth. Intuitively, reduction-in-objective is a metric measuring how effectively the approximated solution $\boldsymbol{X}_{\mathrm{approx}}$ minimizes the original objective.

Table 3: Reduction in objective of different with respect to $k$ large positive eigenvalues provided. Settings follow Figure 4.

|        | k=100 | k=300 | k=500 | k=1000 | k=2000 | k=3000 |
|--------|-------|-------|-------|--------|--------|--------|
| No Fit | 2292  | 1976  | 1733  | 1380   | 501    | 363    |
| Fit    | 122   | 5.59  | 479   | 20     | 140    | 285    |

## D  Experiment Details and Results

We use PyTorch (Paszke et al., 2019) as the deep learning framework and datasets and models provided by its `torchvision` package (if exists). We implement DP-MF training as a new optimizer under Opacus (Yousefpour et al., 2021), a library that enables training PyTorch models with differential privacy, which is also used for our privacy accounting. Our implementation, especially the online noise vector generation, is highly inspired by Granqvist et al. (2024).

All the experiments in this paper are runnable on one single NVIDIA A5000 24GB GPU.

## D.1 EXPERIMENTS SETUPS

**Datasets** We use a variety of datasets, including SVHN (Netzer et al., 2011), CIFAR-10 (Krizhevsky et al., 2009) as two canonical vision tasks. We also evaluate our methods on one medical image dataset, ChestX-ray14 (Wang et al., 2017), as closer to privacy-sensitive applications. When using ChestX-ray14, we perform a multi-label classification, where each image may be annotated with up to 14 thoracic disease categories (e.g., pneumonia, effusion, cardiomegaly, mass). Due to its class imbalance, we compute Area Under the ROC Curve (AUC) for each class and use the average of them (mean AUC) as the accuracy metric. Unless otherwise stated, we use 6,000 TinyImageNet (Le & Yang, 2015) data samples as the public dataset to pretrain the models (Figures 1, 2, 3, and 4, and the main results). We also use those data samples to estimate Hessian/eigenvalues of Hessian for our Problem P1 unless specified otherwise. Figure 3 was further finetuned with CIFAR-10 training data, and Figures 3 and 4 used CIFAR-10 validation set for the Hessian estimation.

**Network Architectures** Our evaluation use both CNN-based (e.g., ResNet152 (He et al., 2016), VGG (Simonyan & Zisserman, 2015)) and Transformer-based (ViTs (Dosovitskiy et al., 2020)) architectures. For some experiments, we use small-scale CNNs (Table 4), whose full Hessian can be computed on our computers. When using a small CNN, except for Figure 2 where we use both, we use CNN with GroupNorm (Table 4a), i.e., in Figures 3 and 4 and Table 1b. For ResNet152 and ViT (vit_b_16), we use the models (together with pretrained weights) provided by torchvision. For VGG, we use the network architecture from Choquette-Choo et al. (2023b), which is smaller than a standard VGG.

Table 4: Network architectures of two small-scale CNN models.

(a) CNN with GroupNorm

| Layer | Parameters |
|---|---|
| Conv2d + GroupNorm + ReLU | $3 \to 16$, kernel $3 \times 3$, stride 1 |
| Conv2d + GroupNorm + ReLU | $16 \to 32$, kernel $3 \times 3$, stride 2 |
| Conv2d + GroupNorm + ReLU | $32 \to 64$, kernel $3 \times 3$, stride 2 |
| AdaptiveAvgPool2d | output $2 \times 2$ |
| Linear + ReLU | $256 \to$ dense_size |
| Linear | dense_size $\to$ num_classes |

(b) CNN with Maxpool

| Layer | Parameters |
|---|---|
| Conv2d + ReLU + MaxPool2d | $3 \to 16$, kernel $3 \times 3$, stride 1; pool $2 \times 2$ |
| Conv2d + ReLU + MaxPool2d | $16 \to 32$, kernel $3 \times 3$, stride 2; pool $2 \times 2$ |
| Conv2d + ReLU | $32 \to 64$, kernel $3 \times 3$, stride 2 |
| AdaptiveAvgPool2d | output $2 \times 2$ |
| Linear + ReLU | $256 \to$ dense_size |
| Linear | dense_size $\to$ num_classes |

**Training/Finetuning Strategy** We use full parameter training (for small CNNs and VGG), parameter-efficient fine-tuning mechanisms including freezing layer (for finetuning the last layer of ResNet152) and LoRA (Hu et al., 2022) (for ViT). For fine-tuning ViTs with LoRA, we use low rank $r = 8$, making the total number of finetuning parameters around 1M. Based on the recommendations of Gu et al. (2025), we finetune those parameters on public data first, then perform privacy-preserving training/finetuning on the private target datasets.

**Hyperparameters** Unless otherwise stated, we split 10% of data samples from training sets as validation sets and **tune hyperparameters on validation sets**. All models are trained for 20 epochs on CIFAR-10 with batch size 450 and 5 epochs on ChestX-ray with batch size 150. We use constant learning rate, no momentum and no weight decay. We use Optuna (Akiba et al., 2019) for auto hyperparameter tuning and search within $[0.1, 1.]$ for the learning rate and $[0.1, 10.]$ for clip norm. Additionally, for DP-BANDMF and NoiseCurve, we search within $[1, 20]$ for the band size.

## D.2 WHAT IF HYPERPARAMETERS ARE TUNED ON TEST SET (A COMMON PRACTICE IN THE DP LITERATURE)

We re-evaluate the experiments in Figure 1 by training the model on the **full training set** and **tuning hyperparameters on the test set**. Under this setup, we are able to reproduce results comparable to those reported in Choquette-Choo et al. (2023a), and our method even achieves higher accuracy. Nevertheless, we emphasize that this is a bad practice and should not be adopted for fair comparison—even in noise-free settings.

Table 5: Test accuracy comparison on CIFAR-10 under various $\epsilon$ ($\delta = 10^{-5}$). We use the **full** train set for training data and tune hyperparameters on test set. Other setups follow Figure 1.

| Non-private | $\epsilon$ | DP-SGD | DP-BANDMF | Our |
|---|---|---|---|---|
| | 1 | 54.68 | 54.68 | **55.99** |
| 82.68 | 2 | 59.01 | 59.01 | **61.12** |
| | 5 | 63.39 | 65.59 | **68.2** |
| | 8 | 65.06 | 67.19 | **72.01** |

### D.3 IN-DISTRIBUTION PUBLIC DATA

In addition to using TinyImageNet as public data for eigenvalue estimation, we also evaluate accuracy when eigenvalues are estimated from the validation set of the target dataset, which can be regarded as in-distribution data relative to the training set.

Table 6: Comparison of test accuracy of using TinyImagenet vs. Validation set as public data under various $\epsilon$ ($\delta = 10^{-5}$). CIFAR-10 experiments following settings of Figure 1 and ChestX-ray experiments follow Table 2.

(a) CIFAR-10

| Non-private | $\epsilon$ | DP-SGD | DP-BANDMF | Tiny | Val |
|---|---|---|---|---|---|
| 84.05 | 5 | 62.17 | 63.29 | 65.77 | 65.91 |
| | 8 | 64 | 64.46 | 69.91 | 70.1 |

(b) ChestX-ray14

| Non-private | $\epsilon$ | DP-SGD | DP-BANDMF | Tiny | Val |
|---|---|---|---|---|---|
| 73.63 | 5 | 58.39 | 59.71 | 62.67 | 64.14 |
| | 8 | 59.66 | 61.99 | 64.28 | 65.9 |

We observe that using validation data eigenvalues yields little-to-no improvement on CIFAR-10 and a modest gain of 1–2% on ChestX-ray14. These results indicate the robustness of our method: even when only out-of-distribution public data such as TinyImageNet are available, the accuracy improvement remains.

### D.4 WHAT IF NOT PRETRAIN

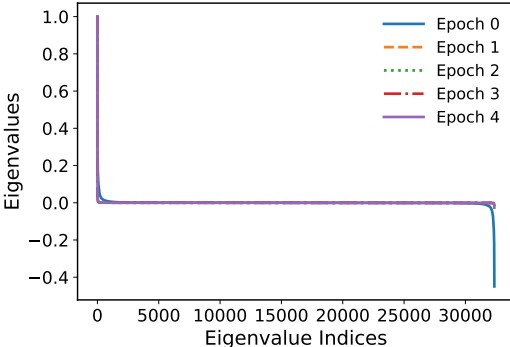

Figure 5: Change of the Hessian eigenvalues during training, a full display of Figure 3a. Despite significant negative eigenvalues at initialization, positive eigenvalues dominate.

We further evaluate the setting without pretraining, where eigenvalues are estimated under the default initialization of the deep learning framework.

Table 7: A comparison of test accuracy of VGG on CIFAR-10 with vs. without pretrain. Other setups follow Figure 1.

| $\epsilon$ | | DP-SGD | DP-BANDMF | NoiseCurve |
|---|---|---|---|---|
| 5 | w/o pretrain | 62.21 | 63.18 | 64.41 |
| | w/ pretrain | 62.17 | 63.29 | 65.77 |
| 8 | w/o pretrain | 64.71 | 65.89 | 68.15 |
| | w/ pretrain | 64 | 64.46 | 69.91 |

The results show that even without pretraining, our method achieves a 1–2% accuracy improvement over DP-SGD and DP-BANDMF. We conjecture that this is because modern initialization schemes (e.g., He (He et al., 2015) or Xavier (Glorot & Bengio, 2010)) are designed to preserve the variance of activations and gradients across layers, which helps maintain stable signal propagation at the start of training and mitigates the risk of the parameters being placed in regions with pathological curvature (see Figure 5). Nevertheless, pretraining remains advantageous: it stabilizes the Hessian spectrum, yielding a nearly constant curvature that strengthens the effectiveness of our method and thereby leads to larger accuracy improvements.

