# OpenReview forum: "Correlating Cross-Iteration Noise for DP-SGD using Model Curvature"
_ICLR.cc/2026/Conference — Submitted to ICLR 2026_

### Official Review · Reviewer_A1NK · 2025-10-16

**Soundness:** 2
**Presentation:** 2
**Contribution:** 2
**Rating:** 4
**Confidence:** 3

**Summary:**

This paper proposes using public data to estimate the Hessian's eigenvalues for improving DP matrix factorization.

**Strengths:**

The experimental results demonstrate that the proposed method can improve utility over DP-SGD and DP-BANDMF.

**Weaknesses:**

1. The core solutions (S1-S4) are presented as empirical observations without theoretical justification. It is unclear why these specific choices work or how they relate to the underlying theory of DP optimization.

2. The proposed algorithm is not formalized with clear, step-by-step pseudocode. This makes it difficult to understand the exact procedure.

**Questions:**

1. The concept of a "model Hessian" is central but not well-explained. If it's data independent, why and how to compute it? If it's data dependent, how does it generalize to the private data distribution using public data?

2. Why does using random labels for the Hessian calculation work? Intuitively, this should produce meaningless curvature information.

3. For the quadratic loss example (line 202), the variable d is not clear. Is this the label? How to understand it when adding H in the middle?

---

> ### Author Response · Authors · 2025-11-24
>
> Thank you for the constructive feedback! Below, we address your specific concerns.
>
> > The core solutions (S1-S4) are presented as empirical observations without theoretical justification. It is unclear why these specific choices work or how they relate to the underlying theory of DP optimization.
>
> We agree that this is an important question for understanding the method more deeply. Our current work does not provide a full, end-to-end theoretical analysis of how the specific approximations in S1–S4 affect non-convex DP training with clipping + noise; developing such a theory would be substantial work, and, to our knowledge, similar non-convex justifications are also not available for the concrete instantiations used in DP-BANDMF. Fully quantifying how each approximation step changes the non-convex DP dynamics would indeed be valuable, and we will provide an update if we finalize a convergence proof during the rebuttal period.
>
> > The proposed algorithm is not formalized with clear, step-by-step pseudocode. This makes it difficult to understand the exact procedure.
>
> We agree that having explicit pseudocode would make the procedure easier to follow. Algorithmically, NoiseCurve uses exactly the same DP-MF / DP-BANDMF training loop once the mixing matrix is fixed: clipping, adding correlated Gaussian noise, subsampling, and accounting are unchanged. These steps are outlined in the paragraph before Section 4. The only additional steps are (i) an offline curvature estimation on public data and (ii) solving our curvature-aware objective to obtain the mixing matrix, after which training proceeds identically to DP-BANDMF. In the revision, we will add a short algorithm box that (a) presents NoiseCurve as a step-by-step procedure, and (b) explicitly highlights where it differs from DP-BANDMF.
>
> > The concept of a "model Hessian" is central but not well-explained. If it's data independent, why and how to compute it? If it's data dependent, how does it generalize to the private data distribution using public data?
>
> Thank you for pointing out this ambiguity. By “model Hessian” we meant the standard, data-dependent Hessian of the loss: $H(w_t) = \frac{1}{n}\sum_{i=1}^n \nabla^2\ell(f(w_t; x_i); y_i)$ . We estimate approximations to the Hessian on public data. We do not construct or assume a separate “data-independent” Hessian anywhere in the method.

---

> ### Author Response · Authors · 2025-11-25
>
> > Why does using random labels for the Hessian calculation work? Intuitively, this should produce meaningless curvature information.
>
> It is fairly common for a Hessian of loss to be unrelated to the labels or to have only a small dependence on the labels. Here are some examples:
>
> - Linear regression: suppose X is a matrix whose rows are feature vectors, y is a vector of labels, and w is the parameter vector. The objective function is $L(w) = ||Xw - y||^2 / 2$.   The gradient is $X^T X w - X^T y$ and the Hessian is $X^T X$. It depends only on the feature vectors and not on the labels.
> - Logistic regression with negative loglikelihood loss (binary cross entropy): Let  $x_1, x_2, ..., x_n$ be the feature vectors, let$y_1, ..., y_n$be their corresponding labels (each $y_i$is either 0 or 1) and let $w$be the weight vector and $\sigma$ the sigmoid function. Then the loss is $L(w)=\sum_{i=1}^n -y_i \log(\sigma(w\cdot x_i)) - (1-y_i)\log(1-\sigma(w\cdot x_i))$. The derivative of the loss function with respect to one parameter $w[a]$ is $\sum_{i=1}^n -y_i(1-\sigma(w\cdot x_i))x_i[a] + (1-y_i)\sigma(w\cdot x_i)x_i[a]$ and the derivative of that with respect to one parameter $w[b]$ is $\sum_{i=1}^n y_i(1-\sigma(w\cdot x_i))\sigma(w\cdot x_i)x_i[a]x_i[b] + (1-y_i)\sigma(w\cdot x_i)(1-\sigma(w\cdot x_i))x_i[a]x_i[b]$ which equals: $\sum_{i=1}^n \sigma(w\cdot x_i)(1-\sigma(w\cdot x_i))x_i[a]x_i[b]$ and this only depends on the feature vectors but not the labels.
> - Deep learning: applying Gauss–Newton decomposition[1, Equation 2], Following that reference, the Hessian of the loss can be expressed as the sum of two terms: $H(w) = \frac{1}{n}\sum_{i=1}^n \left[J_i^\top H_i^{(z)}J_i + \sum_k g_{i,k}\frac{\partial^2 z_{i,k}}{\partial w^2}\right]$, where $J_i = \frac{\partial z_i}{\partial \theta}$ is the Jacobian of the outputs with respect to the model parameters. The term $g_i = \frac{\partial \ell}{\partial z_i}$is the gradient of the loss with respect outputs, and the term $H_i^{(z)} = \frac{\partial^2 \ell}{\partial z_i^2}$ is the Hessian of the loss with respect to the model outputs. For the standard likelihood-based losses that are commonly used (MSE, logistic / softmax cross-entropy, etc.), the overall Hessian of the loss with respect to model parameters has some dependence on the labels, but the Hessian of the loss with respect to the model outputs (i.e., the logits) $H_i^{(z)}$ does not depend on the label $y$. The difference between the two is a secondary term which depends on gradients. At some point, the contribution from the gradients  would be dominated by the label independent term (e.g., near convergence and possible earlier like in the linear and logistic regression examples above).
>
> [1] Vardan Papyan. The full spectrum of deepnet hessians at scale: Dynamics with sgd training and sample size, 2019. URL https://arxiv.org/abs/1811.07062.
>
> > For the quadratic loss example (line 202), the variable d is not clear. Is this the label? How to understand it when adding H in the middle?
>
> Yes, it can be considered to be analogous to the label ($H$ and $d$ depend on data and $w$ is the parameter vector). For clarity, we can revise the paper to use the standard linear regression model where $X$ is a matrix whose rows are feature vectors, y is a vector of labels, w is the parameter vector and the objective function is $L(w) = ||Xw - y||^2 / 2$. The benefit is the more familiar notation, but the downside is the formulas look more cluttered (the gradient becomes $X^TXw-X^Ty$ and the Hessian is $X^TX$). Thus the mapping with our example is $H=X^TX$ and $d=(X^TX)^{-1}X^Ty)$ so that the gradients match $H(w-d) = X^TXw-X^Ty$ and the Hessians match ($H=X^TX$). We had set up the examples to use the shorter notation for attempted clarity but this is easy to revise.

---

> > ### Comment · Reviewer_A1NK · 2025-11-26
> > **Comment**
> >
> > Thank you for your comment. I have a follow-up question regarding the random label case.
> > 1. For linear regression, the Hessian depends only on X, which is clear.
> > 2. For logistic regression, I see the Hessian does not explicitly depend on the labels y. However, it is a function of the model parameters w, which are learned from both the features X and the labels y. When the labels are randomized, do these parameters w (and thus the Hessian) retain any meaningful information? This seems analogous to the deep learning case, where the Hessian also depends on the learned parameters.
> >
> > Meanwhile, can you explain why the loss function is $L(w)=\frac12(w-d)^\top H(w-d)$ given that $H=X^\top X$ and $d=(X^\top X)^{-1}X^\top y$

---

> ### Author Response · Authors · 2025-11-27
>
> > Regarding the random label case
> - Thank you for your quick response. This question is indeed very deep. The short answer is that we believe this forces our approximate Hessian to overestimate the true eigenvalues.
> - This kind of approximate Hessian information appears to be useful in geneal. For example, non-convex proofs that use smoothness assumptions are equivalent to saying all eigenvalues are less than the smoothing parameter L, and the learning rate in those proofs is crucially set to $O(1/L)$. On the other hand, underestimating the eigenvalues seems bad -- for example, setting too many eigenvalues to $0$ is equivalent to approximating the original learning problem with something extremely low rank.
> - If we accept that the goal of an approximate Hessian is to overestimate the true eigenvalues, why would this overestimation happen? The intuition for logistic regression is that training with random labels will try to force predictions for a feature vector $x_i$ to be $\sigma(w\cdot x_i)\approx 1/2$ as there is no link between labels and features. If we denote $p_i$ to be the prediction (output of the sigmoid) for feature vector $x_i$ after supervised training and $q_i$ to be the prediction for feature vector $x_i$ after unsupervised (random-label) pre-training, then the supervised Hessian from the previous comment, in matrix form, would be $X^TD X$ where $X$ is the matrix where row $i$ is feature vector $x_i$ and $D$ is the diagonal matrix whose entry in position $i$ is $p_i(1-p_i)$. The unsupervised Hessian would be $X^TVX$ where $V$ is a diagonal matrix with entries $q_i(1-q_i)$.
> - Now, one expects the supervised predictions $p_i$ to be close to $0$ or $1$ so that $p_i(1-p_i)$ is very small, while the unsupervised predictions $q_i$ are close to $1/2$ and so $q_i(1-q_i)$ would be closer to $1/4$. Therefore the difference between the unsupervised Hessian and the supervised Hessian $X^TVX - X^TDX = X^T(V-D)X$ is likely to be positive semi-definite, meaning that the unsupervised Hessian has larger eigenvalues. In our objective function, the distribution of eigenvalues is all that matters and the distribution of the eigenvalues of the unsupervised Hessian would generally be larger.
> - Thus, we believe that all of our design choices are very simple and have a reasonable mathematical basis (rather than being an over-engineered solution), and we believe this kind of approach is in line with deep learning research standards, but we currently do not have a full mathematical non-convex proof of this concept.
>
> > Meanwhile, can you explain why the loss function is $L(w) = \frac{1}{2}(w - d)^\top H (w - d)$ given that $H = X^\top X$ and $d = (X^\top X)^{-1} X^\top y$?
>
> With the substitution $H = X^\top X$ and $d = (X^\top X)^{-1} X^\top y$, the loss functions $L_1(w)=||Xw-y||^2/2=(Xw-y)^T(Xw-y)/2 = \frac{1}{2}(w^TX^TX w - 2 y^TXw+y^Ty)$ and $L_2(w)=(w-d)^TH(w-d)/2= \frac{1}{2}(w^T H w - 2d^THw+d^Td)=w^TX^TXw - 2y^TXw + d^Td$ are identical up to an additive term ($d^Td-y^Ty$) that is unrelated to $w$.  The loss functions $L_1$ and $L_2$ have the same gradient, Hessian, and optimal value. It originally seemed to us that using  $L_2$ would be more explainable because of its shorter formula for Hessian and gradient (in terms of $H$ and $d$), but given the confusion this choice can cause, we plan to update the discussion using $L_1$ instead.

---

### Official Review · Reviewer_4Wfr · 2025-10-31

**Soundness:** 2
**Presentation:** 3
**Contribution:** 2
**Rating:** 4
**Confidence:** 4

**Summary:**

This paper proposes NoiseCurve, a method to improve the DP-SGD by incorporating model curvature information into cross-iteration noise correlation. The authors propose a new objective on top of the DP-MF framework that interacts between model curvature and DP-SGD with noise. The experiments show promising improvements over DP-SGD and DP-BANDMF across multiple vision datasets.

**Strengths:**

1. The observation on how noise propagates through gradient computation points in DP-MF methods (Eq.2 in Section 2) is an important aspect of correlated noise design. The distinction between direct noise effects and Hessian-mediated effects is well-motivated.
2. Theorem-1 shows that under quadratic loss, only Hessian eigenvalues matter for the optimal objective. This reduction from $O(p^2)$ to $O(p)$ parameters theoretically shows that the geometric structure of the loss landscape relevant to correlated noise is fully captured by the eigenspace. It offers to estimate the curvature by computing eigenvalues alone rather than full Hessian matrices.
3. The paper addresses four challenges (S1-S4) in applying curvature-based noise correlation: using public data for eigenvalue estimation, handling changing/negative eigenvalues, and scaling to large models. Each solution is empirically supported.
4. The experiments span multiple datasets (CIFAR-10, ChestX-ray14), architectures (CNN, ResNet, VGG, ViT), training strategies (full training, fine-tuning, LoRA), and privacy budgets ($\varepsilon$), showing consistent trends in performance.

**Weaknesses:**

1. The paper provides no convergence analysis for the non-convex case or approximation guarantees when using estimated eigenvalues. Additionally, there is no analysis to place bounds on the error introduced by setting negative eigenvalues to zero.
2. Section 6 acknowledges "insufficient understanding'' of when public data choice matters. The claim that eigenspectra "look similar'' across datasets (Figure 2) is based only on small CNNs and limited to vision domains, with no experiments on NLP, tabular or structured data to validate generalization.
3. The eigenvalue approximation strategy is undervalidated despite being critical for scalability to large models. The power-law curve-fitting approach (Section 4.4), while orthogonal to the main contribution, lacks principled guidance for selecting hyperparameters $p_+$ and $\mu_{p_+}$, and is only qualitatively validated on small-scale ($\sim$30,000 param) models (Figure 4).
4. The reported improvements seem moderate (1-2\% in many cases) and often fall within/near the standard deviation ranges, such as in Table 1a, where NoiseCurve achieves $76.5 \pm 0.2$ compared to DP-BANDMF's $75.94 \pm 0.04$. Without statistical significance testing, it is unclear whether these gains are meaningful, and some experimental settings show minimal improvements over the DP-BANDMF baseline.
5. The paper does not report any results on computational overhead, including training time, memory usage, and the costs associated with eigenvalue computation. Therefore, it is difficult to assess the practical feasibility and scalability of the proposed method.

**Questions:**

1. Could the authors provide any theoretical analysis regarding the approximation quality when extending from the quadratic case to non-convex settings? Additionally, it would be helpful to understand how performance degrades as public data becomes less private.
2. What is the computational overhead of NoiseCurve compared to baseline methods like DP-BANDMF? This would help better assess the practical feasibility and scalability of the NoiseCurve approach.
3. Were any statistical significance tests conducted to confirm that the performance NoiseCurve is meaningful? Additionally, it would be helpful to understand under which conditions NoiseCurve provides the most substantial benefits over DP-BANDMF.
4. Is there any way to trace the Hessian powers more efficiently using techniques like Hutchinson's estimator or stochastic trace estimation? This might avoid the computational bottleneck of eigenvalue decomposition while still capturing the essential curvature information.
5. Could the authors comment on whether the similar performance holds for other modalities such as NLP or tabular data? Are there any preliminary experiments or theoretical reasoning to support the generalization of this approach beyond computer vision domains?

---

> ### Author Response · Authors · 2025-11-25
>
> We appreciate the constructive and insightful feedback from the reviewer! It is encouraging that most of the contributions we made in this paper are recognized as strengths. Since each weakness is mapped to an actionable question, below we address your specific concerns by answering those questions.
>
> > Could the authors provide any theoretical analysis regarding the approximation quality when extending from the quadratic case to non-convex settings? Additionally, it would be helpful to understand how performance degrades as public data becomes less private.
>
> - **How performance degrades as public data becomes less private**: Please see Appendix D.3 for our ablation study on the quality of the public data. For both the CIFAR-10 and Chest X-Ray dataset applications, we compared the quality of (1) a Hessian obtained directly from a sample of the training data (best-case scenario, but not privacy preserving), to (2) a Hessian derived from TinyImageNet.  TinyImageNet can be considered a 'similar' public dataset when the application is CIFAR-10, but a very out-of-distribution public dataset when the target is Chest X-Ray. Nevertheless, NoiseCurve improved over the baselines of DP-BandMF and DP-SGD.
> - **The approximation quality when extending from the quadratic case to non-convex settings**: Thank you for the suggestion of adding a convergence proof. Since the prior dominant objective function for DP-BandMF is also missing a non-convex analysis, adding such a result here would indeed have major value. We will provide an update if we finalize a convergence proof during the rebuttal period.
>
> > What is the computational overhead of NoiseCurve compared to baseline methods like DP-BANDMF? This would help better assess the practical feasibility and scalability of the NoiseCurve approach.
>
> Conceptually, NoiseCurve shares the same online training pipeline as DP-BANDMF: once the mixing matrix is computed, the per-iteration correlated-noise generation and DP-SGD updates are identical. The only extra cost is a one-time Hessian eigenvalue estimation on public data (via standard Lanczos-style routines), which is run once per model/setting and then reused across multiple training jobs. In our experiments on a single NVIDIA A5000 GPU, this step did not become a bottleneck compared to DP-SGD training; for example, computing the top 1000 positive eigenvalues for VGG (~550K parameters, used in experiments presented Table 1) takes about 1 hour. In the revision, we will add a small table reporting wall-clock time and peak memory for DP-SGD, DP-BANDMF, and NoiseCurve to make this overhead explicit.
>
> > Is there any way to trace the Hessian powers more efficiently using techniques like Hutchinson's estimator or stochastic trace estimation? This might avoid the computational bottleneck of eigenvalue decomposition while still capturing the essential curvature information.
>
> This is an excellent suggestion. We are indeed exploring Hutchinson-style and related stochastic trace estimation methods as a way to reduce the curvature-estimation bottleneck, and we see this as a promising direction to further improve NoiseCurve’s scalability. However, these experiments are still in progress, and we do not yet have results that we can reliably report.
>
> > Could the authors comment on whether the similar performance holds for other modalities such as NLP or tabular data? Are there any preliminary experiments or theoretical reasoning to support the generalization of this approach beyond computer vision domains?
>
> We agree that this is an important question for assessing the broader applicability of our method. At the moment, our empirical results in the paper are limited to vision tasks, and we do not yet have completed experiments on NLP or tabular benchmarks. We are currently implementing NoiseCurve for transformer-based NLP models to explore this question, but these experiments are still in progress and not yet ready to report.

---

> ### Author Response · Authors · 2025-11-25
>
> > Were any statistical significance tests conducted to confirm that the performance NoiseCurve is meaningful? Additionally, it would be helpful to understand under which conditions NoiseCurve provides the most substantial benefits over DP-BANDMF.
>
> Here are updated versions of Figure 1 (full training a VGG on CIFAR-10) using 6 seeds. It is important to note that for epsilons of 2, 5, 8, every run of NoiseCurve had higher accuracy than every run of DP-BandMF (for the much noisier setting of epsilon=1, only 4 out of the top 6 scores belonged to NoiseCurve). Statistical significance can be assessed in two ways. The first is the 1-sided paired t-test. It is an asymptotic but commonly used test. The second is a permutation test which is non-asymptotic and makes no distributional assumptions (it is equivalent to asking the question: "if there was no difference between the two methods, what is the probability that one of them will always produce higher scores by random chance").
>
> | ε |    DP-SGD    |   DP-BandMF  |  NoiseCurve  |
> |:-:|:------------:|:------------:|:------------:|
> | 1 | 54.04 ± 1.10 | 54.20 ± 1.10 | 55.02 ± 1.54 |
> | 2 | 57.56 ± 0.35 | 58.16 ± 0.27 | 59.85 ± 0.64 |
> | 5 | 63.07 ± 0.52 | 63.08 ± 1.08 | 65.80 ± 0.06 |
> | 8 | 64.55 ± 0.35 | 65.05 ± 0.37 | 68.82 ± 0.16 |
>
> We performed both statistical significance tests (one-sided paired t-test with H1 "our method better than DP-BANDMF", and the permutation test) for the rows in in the table above with the following results:
>
> - $\epsilon=1$: t = 1.69, p = 0.076 (not significant -- the differences are too small to be detectable with 6 seeds).
> - $\epsilon=2$: t = 8.30, p < 0.001 for the one-sided paired t-test. For the permutation test of comparing NoiseCurve (6 seeds) to DP-BandMF (6 seeds), there are $12!/(6! 6!)=924$ orderings of their results. Thus, under the null hypothesis of no difference between the two methods, the probability that every run of NoiseCurve would have the higher accuracy is 1/924 for a p-value of approximately p=0.001.
> - $\epsilon=5$: t = 6.32, p < 0.001 for the one-sided paired t-test. The same analysis as for the epsilon=2 case yields p=0.001 for the permutation test.
> - $\epsilon=8$: t = 19.49, p < 0.00001 for the one-sided paired t-test. The same analysis as for the epsilon=2 case yields p=0.001 for the permutation test.
>
> Table 2 (ViT-LoRA finetuning on ChestX-Ray), with much larger models, is extremely computationally expensive to run. For each seed, we perform hyperparameter tuning for each competitor. Adding 2 seeds takes 1 week using 4 GPU (this is model training time due to the extensive hyperparameter tuning, not Hessian estimation time). Thus, currently we have 3 seeds each for epsilon=5 and 8. Even with 3 seeds, statistical significance can still be assessed by the permutation test, since, for each epsilon, every run of NoiseCurve had higher accuracy than every run of DP-BandMF. The analysis is below the table.
>
> | ε |    DP-SGD    |   DP-BandMF  |  NoiseCurve  |
> |:-:|:------------:|:------------:|:------------:|
> | 5 | 58.02 ± 0.56 | 59.25 ± 0.33 | 62.56 ± 0.49 |
> | 8 | 59.28 ± 0.31 | 62.53 ± 0.39 | 64.58 ± 0.61 |
>
> Combined analysis for $\epsilon=5$ and $\epsilon=8$: For the permutation test of comparing NoiseCurve (3 seeds) to DP-BandMF (3 seeds) there are $6!/(3! 3!)=20$ orderings of their results for $\epsilon=5$ and also 20 orderings for $\epsilon=8$. Thus, under the null hypothesis of no difference between the two methods, the probability that, for each epsilon, every run of NoiseCurve has higher accuracy is $1/20 * 1/20 = 1/400$ resulting in a p-value of $p=0.0025$. We are continuing to run additional experiments to be able to provide per-row significance levels and will provide any updates during the rebuttal period.

---

### Official Review · Reviewer_eVpo · 2025-11-07

**Soundness:** 2
**Presentation:** 2
**Contribution:** 2
**Rating:** 4
**Confidence:** 3

**Summary:**

To resolve the problem of noise addition in Differentially private stochastic gradient descent (DP-SGD) scenarios, NoiseCurve is proposed in this work, that uses model curvature, estimated frompublic unlabeled data, to improve the quality of this cross-iteration noise correlation. The experiments on various datasets, models, and privacy parameters show that the noise correlations computed by NoiseCurve offer consistent and significant improvements in accuracy over the correlation scheme used by DP-MF.

**Strengths:**

- formulates the ojjective for the interaction between model curvature and differentially private SGD with correlated noise.
- obtaining curvature information from unlabeled public data, and coping with the fact that neural network Hessians change during training and have negative eigenvalues.
- the scalability is verified and a method integrating eigenvalue estimation with implicit estimation of the size of the subspace the gradients is presented.

**Weaknesses:**

- Heavily heuristic core objective. The main objective is derived for a quadratic loss with a fixed, known Hessian and then applied to deep, nonconvex networks via proxies; there’s no bound quantifying the gap between the quadratic surrogate and real training dynamics under clipping + DP noise.
- Eigenvalue approximation by curve-fitting lacks guarantees.
- Small-sample reporting. Many figures/tables average over only 3 runs; several plots lack error bars or confidence intervals, making it hard to assess robustness and variance under DP randomness.

**Questions:**

- Multiple runs on the experiments are necessary, such as Table 1.
- More ablation study is necessary, such as, band size, learning rate, clipping norm, noise multiplier, etc.

---

> ### Author Response · Authors · 2025-11-25
>
> Thank you for the constructive feedback! Below, we address your specific concerns.
>
> > Heavily heuristic core objective. The main objective is derived for a quadratic loss with a fixed, known Hessian and then applied to deep, nonconvex networks via proxies; there’s no bound quantifying the gap between the quadratic surrogate and real training dynamics under clipping + DP noise.
>
> Thank you for the suggestion of adding a convergence proof. Since the prior dominant objective function for DP-BandMF is also missing a non-convex analysis, adding such a result here would indeed have major value. We will provide an update if we finalize a convergence proof during the rebuttal period.
>
> > Eigenvalue approximation by curve-fitting lacks guarantees.
>
> We appreciate this concern. Since Hessian eigenvalues depend on data features, there can be no "proof" in the mathematical sense that avoids controversial simplifying assumptions. We do have several kinds of empirical evidence in the paper: it matches the eigenspectrum of models for which it is feasible to compute most eigenvalues (Figure 4). For larger models where direct validation is not possible, the accuracy improvements in Figure 1 and Table 2 provide an indirect justification.
>
> > Small-sample reporting. Many figures/tables average over only 3 runs; several plots lack error bars or confidence intervals, making it hard to assess robustness and variance under DP randomness.
>
> Here are updated versions of Figure 1 (full training a VGG on CIFAR-10) using 6 seeds. It is important to note that for epsilons of 2, 5, 8, every run of NoiseCurve had higher accuracy than every run of DP-BandMF (for the much noisier setting of epsilon=1, only 4 out of the top 6 scores belonged to NoiseCurve). Statistical significance can be assessed in two ways. The first is the 1-sided paired t-test. It is an asymptotic but commonly used test. The second is a permutation test which is non-asymptotic and makes no distributional assumptions (it is equivalent to asking the question: "if there was no difference between the two methods, what is the probability that one of them will always produce higher scores by random chance").
>
> | ε |    DP-SGD    |   DP-BandMF  |  NoiseCurve  |
> |:-:|:------------:|:------------:|:------------:|
> | 1 | 54.04 ± 1.10 | 54.20 ± 1.10 | 55.02 ± 1.54 |
> | 2 | 57.56 ± 0.35 | 58.16 ± 0.27 | 59.85 ± 0.64 |
> | 5 | 63.07 ± 0.52 | 63.08 ± 1.08 | 65.80 ± 0.06 |
> | 8 | 64.55 ± 0.35 | 65.05 ± 0.37 | 68.82 ± 0.16 |
>
> We performed both statistical significance tests (one-sided paired t-test with H1 "our method better than DP-BANDMF", and the permutation test) for the rows in in the table above with the following results:
>
> - $\epsilon=1$: t = 1.69, p = 0.076 (not significant -- the differences are too small to be detectable with 6 seeds).
> - $\epsilon=2$: t = 8.30, p < 0.001 for the one-sided paired t-test. For the permutation test of comparing NoiseCurve (6 seeds) to DP-BandMF (6 seeds), there are $12!/(6! 6!)=924$ orderings of their results. Thus, under the null hypothesis of no difference between the two methods, the probability that every run of NoiseCurve would have the higher accuracy is 1/924 for a p-value of approximately p=0.001.
> - $\epsilon=5$: t = 6.32, p < 0.001 for the one-sided paired t-test. The same analysis as for the epsilon=2 case yields p=0.001 for the permutation test.
> - $\epsilon=8$: t = 19.49, p < 0.00001 for the one-sided paired t-test. The same analysis as for the epsilon=2 case yields p=0.001 for the permutation test.
>
> Table 2 (ViT-LoRA finetuning on ChestX-Ray), with much larger models, is extremely computationally expensive to run. For each seed, we perform hyperparameter tuning for each competitor. Adding 2 seeds takes 1 week using 4 GPU (this is model training time due to the extensive hyperparameter tuning, not Hessian estimation time). Thus, currently we have 3 seeds each for epsilon=5 and 8. Even with 3 seeds, statistical significance can still be assessed by the permutation test, since, for each epsilon, every run of NoiseCurve had higher accuracy than every run of DP-BandMF. The analysis is below the table.
>
> | ε |    DP-SGD    |   DP-BandMF  |  NoiseCurve  |
> |:-:|:------------:|:------------:|:------------:|
> | 5 | 58.02 ± 0.56 | 59.25 ± 0.33 | 62.56 ± 0.49 |
> | 8 | 59.28 ± 0.31 | 62.53 ± 0.39 | 64.58 ± 0.61 |
>
> Combined analysis for $\epsilon=5$ and $\epsilon=8$: For the permutation test of comparing NoiseCurve (3 seeds) to DP-BandMF (3 seeds) there are $6!/(3! 3!)=20$ orderings of their results for $\epsilon=5$ and also 20 orderings for $\epsilon=8$. Thus, under the null hypothesis of no difference between the two methods, the probability that, for each epsilon, every run of NoiseCurve has higher accuracy is $1/20 * 1/20 = 1/400$ resulting in a p-value of $p=0.0025$. We are continuing to run additional experiments to be able to provide per-row significance levels and will provide any updates during the rebuttal period.

---

> ### Author Response · Authors · 2025-11-25
>
> > More ablation study is necessary, such as, band size, learning rate, clipping norm, noise multiplier, etc.
>
> We would like clarification on this question. As described in Appendix D, we performed a hyperparameter search between [0.1, 1] for learning rate, [0.1, 10] for clip norm, and [1, 20] for band size using Optuna, and reported the accuracy from the best-performing hyperparameters for each framework. As it is customary to compare the result from their respective best-performing hyperparameters when comparing two frameworks, we are not sure what the reviewer means by ablation study on learning rate or clipping norm. We do not perform hyperparameter search for noise multipliers as it is not a hyperparameter but is determined given a sampling rate, epsilon, delta, and the number of iterations. It would be appreciated if the reviewer could clarify exactly what kind of ablation studies would be needed.

---

### Meta-Review · Area_Chair_6fPr · 2026-01-06

**Summary:**

The paper proposes NoiseCurve, a method to improve DP-SGD by correlating noise across iterations using curvature (Hessian eigenvalues) estimated from public unlabeled data. The core claim is consistent accuracy gains over DP-SGD and DP-BANDMF across several vision benchmarks. Across reviewers, there is broad agreement that the idea is interesting and empirically promising, but concerns center on: (i) the heavily heuristic nature of the method when extended beyond quadratic losses, (ii) lack of non-convex theory or approximation guarantees, (iii) initially weak statistical validation and small number of runs, and (iv) questions about generality, scalability, and clarity of the algorithmic presentation. All reviewers land slightly below the acceptance threshold but indicate they would not mind acceptance. The rebuttal mainly strengthens the empirical case (more seeds, significance tests, overhead discussion) but does not resolve the theoretical gaps.

**Reviewer Concerns:**

Heuristic objective / lack of non-convex theory (not addressed). All reviewers raised this in some form. The rebuttal explicitly acknowledges the gap and states no non-convex convergence or approximation guarantees are currently available. This remains a core unresolved concern.

Eigenvalue approximation / curvature estimation guarantees (partially addressed): The rebuttal provides intuition, empirical validation on small models, and a detailed discussion of why overestimation may be acceptable (including random-label arguments). Still no formal guarantees; concern remains but is softened.

Small number of runs / lack of statistical significance (largely addressed): The authors added more seeds (where feasible), reported variance, and performed paired t-tests and permutation tests. This directly addresses a major empirical criticism, though some large-model experiments still have only 3 seeds.

Missing ablations (LR, clip norm, band size, etc.) (partially addressed): Authors clarify that hyperparameter searches were already done and explain why some “ablations” may not be meaningful. This resolves confusion but does not add new ablation results.

Algorithm clarity / missing pseudocode (addresssed): Authors acknowledge the issue and commit to adding a clear algorithm box explaining differences from DP-BANDMF. This seems sufficient.

Computational overhead / scalability (addressed): Rebuttal provides concrete wall-clock estimates and explains the one-time cost nature of curvature estimation, with a promise to add a table.

Generality beyond vision (e.g., NLP, tabular) (not addressed): Authors explicitly state no completed experiments beyond vision. This concern remains.

**Reviewer Scores:**

Reviewer eVpo (4 > likely 5): Statistical testing and added seeds address the main experimental concern; theory gap still exists but was already acknowledged.

Reviewer 4Wfr (4 > likely 4): Rebuttal meaningfully improves empirical rigor and clarifies overhead, but non-convex theory and generality concerns remain.

Reviewer A1NK (4 > likeli 4): Detailed responses on Hessian meaning, random labels, and added clarity reduce confusion, but core theoretical objections are unresolved.

---

### Decision · Program_Chairs · 2026-01-26

Reject